# Correlation between *KRAS* Mutation and *CTLA-4* mRNA Expression in Circulating Tumour Cells: Clinical Implications in Colorectal Cancer

**DOI:** 10.3390/genes14091808

**Published:** 2023-09-16

**Authors:** Sharmin Aktar, Farhadul Islam, Tracie Cheng, Sujani Madhurika Kodagoda Gamage, Indra Neil Choudhury, Md Sajedul Islam, Cu Tai Lu, Faysal Bin Hamid, Hirotaka Ishida, Ichiro Abe, Nan Xie, Vinod Gopalan, Alfred K. Lam

**Affiliations:** 1Cancer Molecular Pathology, School of Medicine and Dentistry, Griffith University, Gold Coast, QLD 4222, Australia; sharmin.aktar@griffithuni.edu.au (S.A.); tracie.cheng@griffith.edu.au (T.C.); skodagod@bond.edu.au (S.M.K.G.); mdsajedul.islam@griffithuni.edu.au (M.S.I.); faysal-bin.hamid@alumni.griffithuni.edu.au (F.B.H.); hirotaka_1106@yahoo.co.jp (H.I.); abe1ro@fukuoka-u.ac.jp (I.A.); xienan2@mail.sysu.edu.cn (N.X.); 2Menzies Health Institute Queensland, Griffith University, Gold Coast, QLD 4222, Australia; indraneilchoudhury@gmail.com; 3Department of Biochemistry and Molecular Biology, Mawlana Bhashani Science and Technology University, Tangail 1902, Bangladesh; 4Department of Biochemistry and Molecular Biology, University of Rajshahi, Rajshahi 6205, Bangladesh; farhad_bio83@ru.ac.bd; 5Faculty of Health Sciences & Medicine, Bond University, Gold Coast, QLD 4229, Australia; 6Department of Biochemistry & Biotechnology, University of Barishal, Barishal 8254, Bangladesh; 7Department of Surgery, Gold Coast University Hospital, Gold Coast, QLD 4215, Australia; cutlu01@yahoo.com; 8Pathology Queensland, Gold Coast University Hospital, Southport, QLD 4215, Australia

**Keywords:** circulating tumour cells, immune checkpoint molecules, colorectal carcinoma, *KRAS*, *CTLA-4*

## Abstract

Combination strategies of *KRAS* inhibition with immunotherapy in treating advanced or recurrent colorectal carcinoma (CRC) may need to be assessed in circulating tumour cells (CTCs) to achieve better clinical outcomes. This study aimed to investigate the genomic variations of *KRAS* in CTCs and matched CRC tissues and compared mRNA expression of *KRAS* and *CTLA-4* between wild-type and *KRAS*-mutated CTCs and CRC tissues. Clinicopathological correlations were also compared. Six known mutations of *KRAS* were identified at both codon 12 and codon 13 (c.35G>T/G12V, c.35G>A7/G12D, c.35G>C/G12A, c.34G>A/G12S, c.38G>C/G13A, and c.38G>A/G13D). Three CTC samples harboured the identified mutations (16.7%; 3/18), while fifteen matched primary tumour tissues (65.2%, 15/23) showed the mutations. CTCs harbouring the *KRAS* variant were different from matched CRC tissue. All the mutations were heterozygous. Though insignificant, *CTLA-4* mRNA expression was higher in patients carrying *KRAS* mutations. Patients harbouring *KRAS* mutations in CTCs were more likely to have poorly differentiated tumours (*p* = 0.039) and with lymph node metastasis (*p* = 0.027) and perineural invasion (*p* = 0.014). *KRAS* mutations in CTCs were also significantly correlated with overall pathological stages (*p* = 0.027). These findings imply the genetic basis of *KRAS* with immunotherapeutic target molecules based on a real-time platform. This study also suggests the highly heterogeneous nature of cancer cells, which may facilitate the assessment of clonal dynamics across a single patient’s disease.

## 1. Introduction

Understanding the biology of colorectal carcinoma (CRC) has led to therapeutic breakthroughs, improving patient care. Yet, intratumor heterogeneity challenges single-site biopsies in unveiling genomic landscapes. With emerging targeted therapies, it becomes crucial to screen tumours for genomic changes and understand resistance. It is also important to note the changes in circulating tumour cells (CTCs), which may differ genomically from the primary tumour and could provide a better reflection of the current disease status. In this context, the utilisation of CTCs, which can be captured through a minimally invasive blood test, emerges as a promising avenue to gain valuable insights into intratumor heterogeneity and tumour evolution.

Around 52% of patients with CRCs are accompanied by a mutation in *KRAS (Kirsten rat sarcoma virus)*. In these patients, activating mutations in the oncogene *KRAS* usually occur at exon 2 (codons 12 and 13), and less frequently at exons 3 (codons 59 and 61) and 4 (codons 117 and 146) [1,2]. These mutations predict poor response to standard targeted therapy and anti-EGFR (epidermal growth factor receptor) treatment because they activate the *RAS/MAPK* (mitogen-activated protein kinase) pathway constitutively [3]. However, a large portion of the patients do not carry *KRAS* mutations and the durability of response in patients with *KRAS* mutations may be short due to the development of resistance.

On the other hand, the discovery of immune checkpoint inhibitors, such as anti-PD-1 (programmed cell delath protein 1) and anti-CTLA-4 (cytotoxic T-lymphocyte-associated protein 4, CD152) antibodies, has shown improved patient outcomes for numerous cancers, including CRC [4,5,6]. Altered expressions of *KRAS*, *BRAF*, *p53*, *MYC*, *APC (adenomatous polyposis coli)*, and *PTEN (phosophatase and tensin homolog)* play a role in controlling tumour–immune system crosstalk. Also, they can modulate the expression of immune checkpoint molecules such as *PD-1, PD-L1* (programmed cell death ligand 1), *PD-L2* (programmed cell death ligand 2), *CTLA-4 )*, *CD47*, etc., in several cancers [7,8,9,10,11,12,13,14,15,16,17,18,19]. For instance, loss of *p53* activity increases PD-L1 surface expression, leading to T-cell inactivation [20]. Several oncogenic events, such as *KRAS* mutation or *MYC* activation, have also been shown to suppress or evade anti-tumour immune responses via immune checkpoint molecules [21,22].

For the past few decades, CTCs have been studied extensively because of their potential as a non-invasive liquid biomarker for prognostic and predictive value [23,24]. In addition to cell enumeration, the genetic profiling of CTCs has gained growing interest as a means to investigate intratumor heterogeneity [25,26]. To illustrate, Kalikaki et al. (2014) examined *KRAS* mutations within CTC-enriched samples, revealing the presence of detectable mutations in CTCs compared to matched primary tumours [25]. When evaluating *KRAS* mutations in consecutive blood samples, it was observed that the mutational status of *KRAS* in individual patients’ CTCs varied during the course of treatment. Given the existence of intratumor heterogeneity in various tumour types [27,28,29], an important query emerges concerning whether CTCs possess the capability to mirror the genetic attributes of tumours. To delve into this matter, we performed an investigation into the sequence profiles of CTCs and their corresponding primary tumours. Additionally, we aimed to compare the *KRAS* and *CTLA-4* expression levels with the *KRAS* mutation status in patients with CRCs to characterise the clinical and molecular aspects of *KRAS*-mutated CTCs in patients with CRC to determine whether the mutation affects the expression of *CTLA-4* in CTCs in colon carcinogenesis.

Hence, the genetic profiling of CTCs holds promise as an innovative approach to gain a more comprehensive understanding of the dynamic characteristics of tumours in real-time, providing an alternative to invasive procedures. Furthermore, it could contribute to the development of treatment strategies for CRC patients by elucidating the genetic basis of immunotherapeutic target molecules on a real-time platform. However, the actual clinical utility of CTCs in revealing *KRAS* mutation status along with immunotherapeutic target molecules is yet to be fully determined. 

Our previous study found a positive correlation between *CTLA-4* mRNA and *KRAS* mRNA expressions in CTCs isolated from patients with CRC [30]. This may suggest that the activation of driver genes could regulate immune response through altered immune checkpoint pathways. To delve into this matter, we performed an investigation into the sequence profiles of CTCs and their corresponding primary tumours. Additionally, we aimed to compare the *KRAS* and *CTLA-4* expression level with the *KRAS* mutation status in patients with CRCs to characterise the clinical and molecular aspects of *KRAS*-mutated CTCs in patients with CRC to determine whether mutation affects the expression of *CTLA-4* in CTCs in colon carcinogenesis.

## 2. Materials and Methods

### 2.1. Population and Sample Recruitment

Consecutive patients who underwent resection of CRC by a colorectal surgeon (CTL) at Gold Coast University Hospital, Queensland, Australia, gave consent for the collection of peripheral blood and cancer tissue. From each patient, 10 mL of peripheral blood was freshly collected in heparin-containing BD (Becton Dickinson, Franklin Lakes, NJ, USA) vacutainer tubes at the time of surgery and was processed within one hour of collection. The tissues collected were immediately snap-frozen with liquid nitrogen and stored at −80 °C until use. Ethical approval was obtained from the Griffith University Human Research Ethics Committee (GU Ref No: MSC/17/10/HREC). Before enrolling in the trial, all patients provided written informed consent. A pathologist (A.K.L.) evaluated the pathological characteristics of resected tissues collected from patients with CRC along the lines of the World Health Organisation (WHO) classification of tumours [31]. The clinical and pathological characteristics, including patient age, gender, tumour size, site, histological subtype, microsatellite instability (MSI) status, and pathological staging, were also collected [32]. Among 57 patients, a total of 23 patients positive for CTCs were used in this study. Table 1 demonstrates the demographic characteristics of patients.

### 2.2. CTC Enrichment and Staining

CTCs were enriched from 10 mL of peripheral blood according to the manufacturer’s protocol (EasySep^TM^ Direct Human CTC Enrichment Kit, STEMCELL Technologies, Vancouver, BC, Canada). The procedures of CTC enrichment and isolation as well as the cell spiking method to verify the specificity and sensitivity of this method have been previously reported in detail [30,33]. The enriched cell pellets were seeded in a 96-well plate after centrifugation at 450 rcf for 7 min and incubated at 37 °C for 12 h.

Immunofluorescence staining was performed to identify CTCs as described previously [30]. Mouse anti-EPCAM antibody (Thermo Fisher Scientific, Waltham, MA, USA), goat anti-SNAI1 (Santa Cruz Biotechnology, Paso Robles, CA, USA), mouse anti-E-cadherin (Santa Cruz Biotechnology), and goat anti-MMP9 (C-20) (Santa Cruz Biotechnology) were used as primary antibodies and rabbit-anti-mouse IgG fluorescein isothiocyanate (FITC) and rabbit anti-goat IgG (H + L) Texas Red (Sigma-Aldrich, St. Louis, MO, USA) as secondary antibodies. Hoechst 33342 (Thermo Fisher Scientific, Waltham, MA, USA) was used to stain the nucleus. The stained cells were counted by a Widefield Microscope: Nikon Ti-2 (Nikon Corporation, Tokyo, Japan) at 20× magnification. The possible cross-species binding of selected secondary antibodies was checked to avoid overlapping results as previously published [30].

### 2.3. Extraction of DNA and RNA and cDNA Conversion

The selected frozen tissues were sectioned into 5–6 μm slices using a cryostat (Leica Biosystems, Mt Waverley, VIC, Australia) for haematoxylin and eosin staining. The sections were then examined by an anatomical pathologist (A.K.L.) to confirm that cancer cells made up more than 70% of the volume of the samples. DNA was extracted and purified from selected tissue sections with Qiagen DNeasy Blood & Tissue kits (Qiagen Pty. Ltd., Hilden, Germany) following the manufacturer’s guidelines. RNA was extracted from the fresh frozen tissue sections using the Qiagen miRNeasy Mini Kit. DNA and RNA from CTCs were extracted using the Qiagen AllPrep DNA/RNA Mini Kit. The purity of the extracted DNA and RNA was measured with the optical density method (260/280 ratio) using a nanodrop spectrophotometer (BioLab, Ipswich, MA, USA). The concentrations were noted in ng/μL.

### 2.4. Whole-Genome Amplification (WGA) Reaction

Due to the low number of cells in CTC fractions, DNA isolated from CTC fractions was subsequently amplified according to the manufacturer’s protocol using the REPLI-g Advanced DNA Single Cell Kit (Qiagen Pty. Ltd., Hilden, Germany). Briefly, 15 μL of DNA was incubated with 2 μL of DNA lysis buffer (DLB) for 3 min at room temperature, and then 3 μL of stop solution was added. When the DNA had been denatured, it was added to a master mix consisting of 29 L of reaction buffer and 2 L of REPLI-g sc DNA polymerase. The reaction was incubated for 2 h at 30 °C followed by 3 min at 65 °C. DNA for WGA was stored at −80 °C.

### 2.5. Primer Design

*KRAS* and *CTLA-4* gene expression was analysed using 5ʹ GGCCTGCTGAAAATGACTG 3ʹ forward and 5ʹ CTTGCTTCCTGTAGGAATCCTC 3ʹ reverse, and 5′ TTGCTAAAGAAAAGAAGCCC 3ʹ forward and 5ʹ AAAGTTAGAATTGCCTCAGC 3′ reverse primers, respectively. Primers specific for detecting *KRAS* mutations in exon 2 and exon 3, specifically the common *KRAS* mutations at codon 12, codon 13, codon 59, and codon 61, were designed using Primer3web version 4.1.0 [34]. *KRAS* exon 2 was amplified using 5′ AAGCGTCGATGGAGGAG 3′ forward primer and 5ʹ CGTCAAGGCACTCTTGC 3′ reverse primer. Exon 3 was amplified using 5ʹ TCCAGACTGTGTTTCTCCCTTC 3′ forward and 5′ CAAAGAAAGCCCTCCCCAGT 3ʹ reverse primer. All primers were cross-checked for specificity using Primer Blast (https://www.ncbi.nlm.nih.gov/tools/primer-blast/, accessed on 17 November 2022). The primer pairs were purchased from Sigma-Aldrich (St. Louis, MO, USA) and Integrated DNA Technologies Australia Pty Ltd. (Melbourne, VIC, Australia).

### 2.6. Quantitative Real-Time Polymerase Chain Reaction Analysis

First-strand complementary DNA (cDNA) was synthesised by converting 1 μg of total RNA into cDNA by the SensiFAST cDNA synthesis kit (Meridian Bioscience, Cincinnati, OH, USA) according to the manufacturer’s guidelines. Alterations in KRAS and CTLA-4 gene expressions in CTC fractions and CRC tissue samples were investigated using real-time polymerase chain reaction (RT-PCR) (QuantStudio, Thermo Fisher Scientific, Waltham, MA, USA). Quantitative PCR was performed in a volume of 10 μL reaction mixture consisting of 5 μL of SensiFAST SYBR No-ROX (Bioline, Meridian Bioscience, London, UK), 0.4 μL of 10 μmoL/L forward and reverse primer, 3.2 μL of 0.1% diethylpyrocarbonate (DEPC)-treated water, and 1 μL of cDNA template (100 ng/μL). The housekeeping gene, β-actin, was used as an internal control because of the established consistent results in CRC tissues [35]. The Ct values <35 for all target genes and <30 for housekeeping genes were included in the gene expression analysis. The gene expression analysis was performed as previously reported [30,36,37]. If the fold change value was greater than 1, the gene expression level was categorised as high; conversely, if it was less than 1, the expression level was considered low. Healthy blood samples (*n* = 6) were used as calibrators to exclude the possible leukocyte contamination in CTC fractions, as discussed in the previous reports [30].

### 2.7. High-Resolution Melt (HRM) Curve Analysis

Genomic DNA extracted from CRC tissues and matched CTCs were used for HRM analysis to screen the mutations in the *KRAS* gene at known mutation sites. HRM curve analysis and target gene amplification were performed on QuantStudio 6/7 Flex systems using HRM software version 3.2 (Thermo Fisher, Waltham, MA, USA). Exons 2 and 3 of *KRAS* were amplified using SensiMIX HRM mastermix (Meridian Bioscience, Cincinnati, OH, USA) following the manufacturer’s protocol. Melt curves were recorded for every 0.05 °C/s temperature rise between 65 and 85 °C. When both replicates of HRM analysis showed a variation relative to the wild type, we considered the result as a mutant, as described in previous reports [38,39,40].

### 2.8. Sanger Sequencing

Following HRM, the unpurified products were sent to the Australian Genome Research Facility (AGRF) for sequencing. The unpurified PCR products were first cleaned up from gel and purified within the AGRF facility and then sequenced using the respective forward and reverse primer by the Big Dye Terminator chemistry version 3.1 (Applied Biosystems, Foster City, CA, USA) with standardised cycling PCR conditions and analysed by the 3730xl Capillary sequencer (Applied Biosystems). The DNA sequencing reactions were produced by following the AGRF sample preparation guide. The resultant chromatograms were analysed by a sequence analyser, Chromas 2.5.0 software. The identification of mutations was made by NCBI BLAST: Basic Local Alignment Tool (https://blast.ncbi.nlm.nih.gov/Blast.cgi, accessed on 15 February 2023) and further verified by Indigo: Rapid Indel Discovery in Sanger Chromatograms (https://www.gear-genomics.com/indigo, accessed on 15 February 2023). Both forward and reverse sequences were meticulously analysed to confirm mutations.

### 2.9. Statistical Analysis

Gene expression analysis of *KRAS* and *CTLA-4* in CTCs and matched tumour tissues was performed using GraphPad Prism 7 (San Diego, CA, USA). Comparisons of *KRAS* and *CTLA-4* expressions (fold changes) and *KRAS* mutation status and clinicopathological parameters were performed using the chi-square test, the likelihood ratio, and Fisher’s exact test (Version 29; IBM SPSS Inc., Armonk, NY, USA). The statistical significance level was considered at *p* < 0.05.

## 3. Results

### 3.1. CTCs Identification

A total of 23 out of 57 patients were included in this study who were CTC-positive with a wide range of CTC counts (range: 2–200). Patients positive for all four surface markers (EpCAM, SNAIL1, E-cadherin, and MMP-9) and the number of different subpopulations of CTCs in individual patients are represented in Figure 1.

### 3.2. Identification of KRAS Mutations in CTCs and CRC Tissues

We evaluated the *KRAS* mutations (codons 12 and 13 in exon 2 and codons 59 and 61 in exon 3) in CTC samples obtained from 23 patients with CRC. Then, we matched their results with the results of *KRAS* mutations detected in primary tumours. *KRAS* gene mutations in CTCs and CRC tissue were initially suspected based on the deviated melt curves in HRM analysis (Figure 2). All mutations of *KRAS* in CTC samples were found in both codon 12 and codon 13. No mutations were noted in exon 3. Six known heterozygous mutations were identified: c.35G>T/G12V, c.35G>A7/G12D, c.35G>C/G12A, c.34G>A/G12S, c.38G>C/G13A, and c.38G>A/G13D. Among them, three mutations were identified in the CTCs and fifteen in matched primary tumour tissues (Table 2).

*KRAS* mutations were noted in 13.04% (3/23) of CTCs, whereas 65.2% (15/23) showed *KRAS* mutations in matched primary tumours. Two of the three *KRAS*-mutated CTCs harboured G12D (c.35G>A) mutation, while the other one was detected with G13A (c.38G>C) mutation. In twelve primary tumour samples, a *KRAS* mutation was observed in codon 12; in three samples, the mutation was observed in codon 13. Furthermore, 50% (7/15) of patients showed G12V (c.35G>T) in CRC tissue. No concordance between *KRAS* mutations in CTCs and primary tumours was found.

Additionally, two patients harboured a *KRAS* mutation in CTCs, which is dissimilar to the matched primary tumours. Another patient harboured a mutation in CTCs, but a wild-type *KRAS* was identified in the primary tumour. Table 2 shows the presence of *KRAS* mutations at different positions in CTC samples and matched primary tumours.

### 3.3. Correlation between KRAS and CTLA-4 mRNA Expression

The majority of CTC samples showed high *KRAS* and *CTLA-4* expression, while all the CRCs had low gene expression for these markers (Figure 3A,B). We discovered *KRAS* mutations in CTCs from a small group of only three patients. Hence, we present the levels of *KRAS* and *CTLA-4* mRNA expression in individuals with both wild-type (WT) and mutant (MT) *KRAS* cases, identified in both CTCs and CRC tissues. Most of the cases that harboured *KRAS* mutations either in CTCs or in tumour tissues had higher *KRAS* expression (Figure 3C). Similarly, *CTLA-4* is predominantly expressed higher in CTCs in patients harbouring *KRAS* mutations (Figure 3D).

However, we did not find any correlation between *KRAS* mutation and *KRAS* and *CTLA-4* mRNA expression in CTCs and primary tumours (*KRAS*: *p* = 0.56, *CTLA-4*: *p* = 0.24) (Table 3). Due to the low number of CTCs, protein expression was not feasible to perform on the samples.

### 3.4. Clinicopathological Correlations

The *KRAS* status in CTCs and primary tumours and its association with clinical-pathological parameters are shown in Table 4. All CTCs harbouring *KRAS* mutations were from patients with advanced pathological stages (III or IV) and with lymph node metastasis. *KRAS* mutation in CTCs was also significantly correlated with perineural invasion (5% vs. 66.7%, *p* = 0.014). Though insignificant, all *KRAS* mutant CTC samples showed advanced local spread (T3 or T4) (*p* = 0.09).

No correlation is noted between the *KRAS* mutation status in CRC tissue and clinical parameters. However, of the 15 patients diagnosed with advanced local spread, 11 patients harboured *KRAS* mutations in the primary tumours, though this was not significant (*p* = 0.09). Like CTCs, it is noted that cancers with a microsatellite instability phenotype were more likely not to harbour this mutation in the primary tumours (*p* = 0.06).

The association between clinicopathological parameters and *KRAS* and *CTLA-4* gene expression levels in CTCs was also compared (Table 5 and Table 6). There was a significant association between the *KRAS* expression in CTC fraction and the age of the patients in this cohort. Patients below 60 years old had higher *KRAS* expression than those >60 years old (85.7% vs. 43.8%, *p* = 0.05). Although not statistically significant, *KRAS* was more likely to be expressed highly in the rectum (70% vs. 46.2%), in patients with an advanced tumour (T) stage (60% vs. 50%) and with lymph node metastasis (72.5% vs. 41.7%) and advanced (III or IV) pathological stages (72.5% vs. 41.7%) and with no MSI (10/13, 77%).

On the other hand, we found a significant correlation between the *CTLA-4* mRNA expression level in CTCs and the tumour stage (T) (Table 6). *CTLA-4* is more likely to be positively expressed in patients with advanced T stages (III or IV) (73.3% vs. 37.55, *p* = 0.036). Around 52.2% of patients with moderately differentiated tumours had a positive expression of *CTLA-4* (*p* = 0.01). Like *KRAS*, it is noted that *CTLA-4* expression was also higher in patients with lymph node metastasis, advanced pathological stages (72.7% vs. 50.0%), and MSI-stable cancer (68.4% vs. 25.0%). Also, patients with large tumours (>40 mm) had higher *CTLA-4* expression (70% vs. 53.8%).

## 4. Discussion

Since CTCs can provide real-time information regarding tumour biology, genotyping them can provide a platform for research into cellular heterogeneities, resistance mechanisms, and therapeutic targets in cancer. In this study, we compared the prevalence of *KRAS* mutations in CTCs to that of their corresponding primary tumours from patients with CRC.

We found no concordance of *KRAS* status between CTC and primary tumours, which is consistent with a previous report [41]. However, few previous studies found higher concordance between CTCs and primary tumours [42,43,44,45]. This discrepancy may happen because the CTC subclone that is shed into the bloodstream might be genetically distinct from the primary tumour sections that were analysed. Several studies have also attempted to account for the fact that CTCs harbouring *KRAS* mutations are dissimilar to their matched primary tumours [26,45,46,47]. Intratumor heterogeneity is one of the reasons for this discordance, suggesting that both *KRAS* wild-type and mutant subpopulations of cells may coexist within the same tumour and compete with one another for shedding into the bloodstream. Another reason is that as tumours evolve, CTCs may acquire unique mutations, which could explain why mutations exist in CTCs but not in the primary tumours. Despite the low sample size, these findings could imply that *KRAS* wild-type CTCs are often found in the peripheral blood of patients harbouring mutant primary tumours.

*KRAS* mutant status was associated with unique clinical–pathological features in patients with CRC. Most mutations in the current study, in both CTCs and CRC tissues, were from advanced local tumour spread and pathological stages and showed a higher rate of lymph nodes with metastatic carcinoma. These findings are consistent with previous findings that patients with *KRAS* mutations are more likely to be aggressive, which promotes tumour progression [48,49]. Previous research also supports the hypothesis that *KRAS* mutations are more common in patients exhibiting the microsatellite stability phenotype [48,50]. Perineural invasion (PNI) is associated with a poorer prognosis, because neoplastic cells positioned along nerve fascicles are difficult to remove during radical surgery, leading to disease recurrence [51,52,53]. In the current study, almost all patients with perineural invasion had *KRAS* mutations, suggesting that *KRAS* mutation status can predict tumour growth and progression as well as recurrence, though we did not find any correlation between *KRAS* status and disease recurrence. The lack of association between CTCs and primary tumours may be due to the low frequency of mutations and the small number of cases analysed.

The present work also investigated *KRAS* and *CTLA-4* mRNA expression in CTCs and matched primary tumour tissues. A correlation between these two genes suggests a potential role of *CTLA-4* in the *KRAS*-mediated carcinogenic pathway. It is worth noting that high expression of *CTLA-4* is observed in most *KRAS*-mutated CTCs and CRC tissues. Additionally, higher *CTLA-4* expression was more likely to be aggressive, to be of higher histological grade and advanced tumour stages, as well as advanced pathological stages. These results suggest the cancer-promoting function of *CTLA-4* in CRC through immune escape pathways, thus facilitating tumour spread.

However, different CTC subpopulations and the usage of CTC fractions rather than a single CTC, which might be contaminated with diverse cell populations such as leukocytes or normal epithelial cells due to the limitation of the CTC isolation procedure, may confound the conclusions of the current experiment. Molecular characterisation of CTCs at the single-cell level would avoid this limitation. To address this problem, three blood samples from patients negative for CTCs were analysed for *KRAS* mutations, as discussed in a previous report [42]. Although no mutations were found, PCR amplification detected the presence of haematopoietic cells in those samples. Nonetheless, it is suggested that the constraints inherent in the process of amplification and sequencing could potentially contribute to the observed heterogeneity [26]. Moreover, the utilisation of distinct sequencing techniques could potentially lead to disparate outcomes in terms of mutations detected. In a comparative study involving high-resolution melt (HRM), allele-specific PCR (ASPCR), and pyrosequencing methods, Suhaimi et al. observed discrepancies in the mutation status determined through the aforementioned approaches [47].

In conclusion, given the inherent restrictions of current molecular biomarkers due to intratumor heterogeneity, these results may facilitate the assessment of clonal dynamics in a single individual patient. However, due to the limited number of *KRAS* mutations detected in CTCs, we could not show the correlation between *CTLA-4* expression and *KRAS* mutations in CTCs. More research is warranted to determine the clinical significance of genomic profiling of CTCs during *KRAS*-mediated CRC carcinogenesis in predicting responsiveness to anti-CTLA-4 targeted therapy.

## Figures and Tables

**Figure 1 genes-14-01808-f001:**
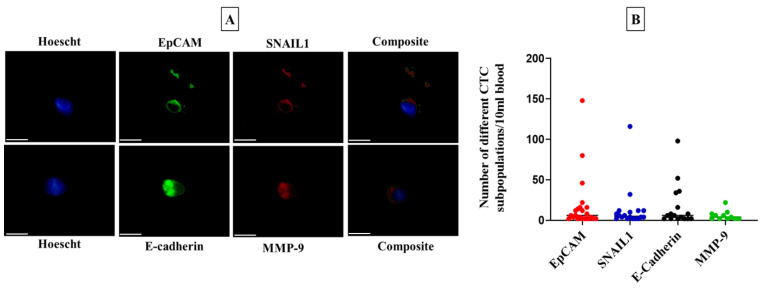
Enumeration of circulating tumour cells (CTCs) in colorectal carcinoma patients. (**A**) A representative image of CTCs detected from patients with colorectal carcinoma (Scale bar: 50 µm), (**B**) a comparison of the number of different subpopulations of CTCs detected in patients with colorectal carcinoma.

**Figure 2 genes-14-01808-f002:**
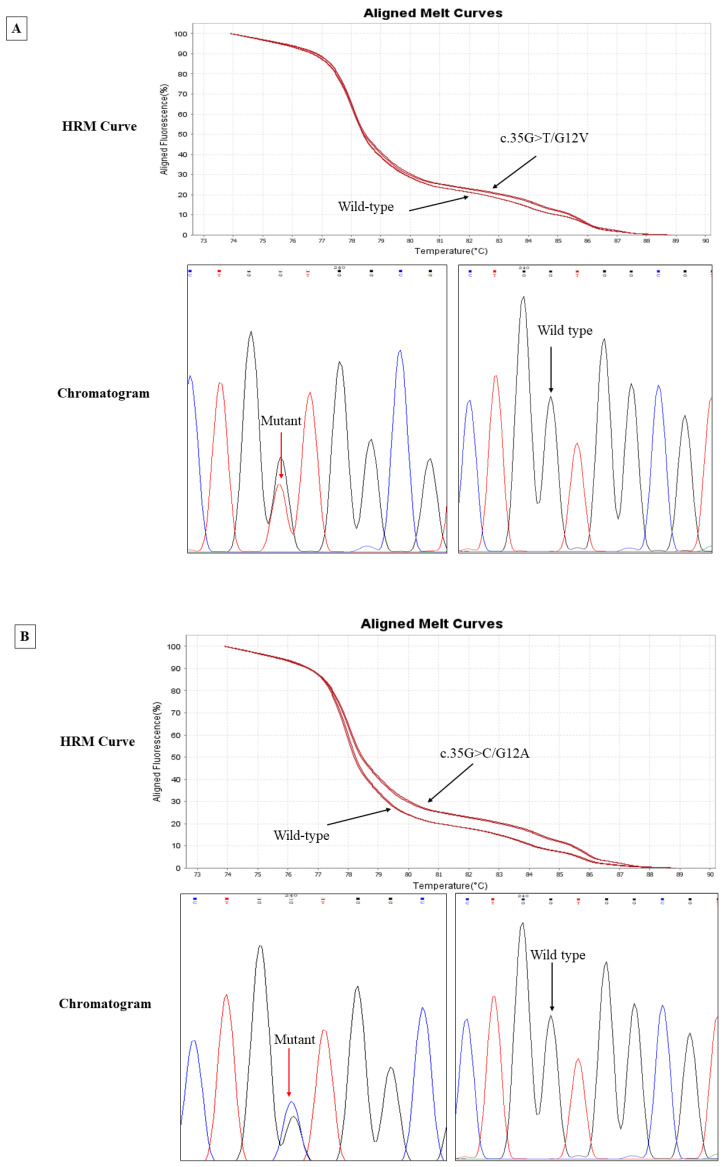
Exonic mutations were detected in CTCs and matched CRC tissues. Representative wild-type and mutant sequences of *KRAS* as seen via high-resolution melt (HRM) versus Sanger sequencing analysis in exon 2 and exon 3 are represented. Black arrow in HRM curve represents the heterozygous variant deviation against wild type. Different colured lines in chromatogram indicates the presence of six known heterozygous mutations, (**A**) c.35G>T/G12V, (**B**) c.35G>A7/G12D, (**C**) c.35G>C/G12A, (**D**) c.34G>A/G12S, (**E**) c.38G>C/G13A, and (**F**) c.38G>A/G13D against the respective wild type.

**Figure 3 genes-14-01808-f003:**
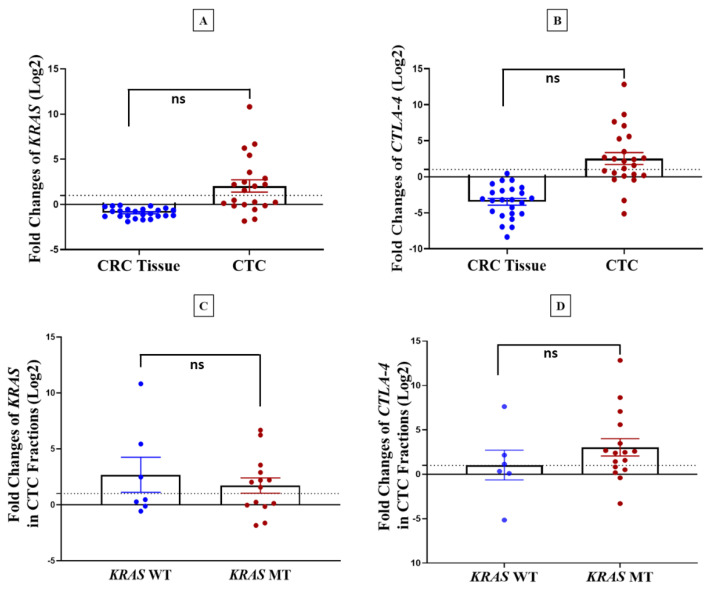
*KRAS* and *CTLA-4* mRNA expression level in CTCs and CRC tissues and their comparison with KRAS mutation status in patients with CRC. (**A**,**B**) Fold change (log2) expression of *KRAS* and *CTLA-4* in CTCs calculated relative to healthy donor blood samples and in CRC tissues calculated relative to the expression in adjacent non-neoplastic mucosa tissue and normalised by β-actin as an internal control. (**C**,**D**) Relative fold change expression level of *KRAS* and *CTLA-4* in *KRAS* wild type *(KRAS* WT) and *KRAS* mutant (*KRAS* MT) cases detected in both CTC fractions and CRC tissues. The dashed line indicates the normal fold change value. ns; not significant.

**Table 1 genes-14-01808-t001:** Clinical and pathological characteristics of patients with colorectal cancer in this series.

Characteristics	Total (23)
**Gender**	
Female	13 (56.5%)
Male	10 (43.5%)
**Age**	
≤60 years	7 (30.4%)
>60 years	16 (69.6%)
**Size**	
≤40 mm	13 (56.5%)
>40 mm	10 (43.5%)
**Site**	
Colon	13 (56.5%)
Rectum	10 (43.5%)
**Grade**	
Well (1)	4 (17.4%)
Moderate (2)	16 (69.6%)
Poor (3)	3 (13.0%)
**T Stage**	
I or II	8 (34.8%)
III or IV	15 (65.2%)
**Lymph node status**	
Negative	12 (52.2%)
Positive	11 (47.8%)
**Distant metastasis**	
Negative	21 (91.3%)
Positive	2 (8.7%)
**Overall pathological stage**	
I or II	12 (52.2%)
III or IV	11 (47.8%)
**Microsatellite instability (MSI)**	
Stable	19 (82.6%)
High	4 (17.4%)

**Table 2 genes-14-01808-t002:** Exon mutations of *KRAS* at codon 12 and codon 13 detected in circulating tumour cells (CTCs) and matched colorectal cancer (CRC) tissues.

Patient ID	# of CTCs	*KRAS* Mutation	
		CTC	Tumour
Patient 1	32	WT	c.35G>T/G12V
Patient 2	12	c.35G>A7G12D	WT
Patient 3	14	WT	c.35G>T/G12V
Patient 4	4	WT	WT
Patient 5	2	WT	c.34G>A/G12S
Patient 6	2	WT	c.34G>A/G12S
Patient 7	6	WT	c.35G>T/G12V
Patient 8	2	WT	c.35G>T/G12V
Patient 9	6	c.35G>A7G12D	c.35G>T/G12V
Patient 10	4	c.38G>C/G13A	c.35G>T/G12V
Patient 11	16	WT	c.38G>A/G13D
Patient 12	4	WT	WT
Patient 13	10	WT	WT
Patient 14	6	WT	WT
Patient 15	52	WT	c.35G>T/G12V
Patient 16	80	WT	WT
Patient 17	30	WT	WT
Patient 18	200	WT	c.35G>C/G12A
Patient 19	20	WT	c.35G>A7/G12D
Patient 20	6	WT	WT
Patient 21	4	WT	c.34G>A/G12S
Patient 22	100	WT	c.38G>A/G13D
Patient 23	80	WT	c.38G>A/G13D

WT: wild type.

**Table 3 genes-14-01808-t003:** Correlation of *KRAS* and *CTLA-4* mRNA expression level in CTC fractions with the *KRAS* mutation status either in CTCs or in CRC tissues.

*KRAS* Mutation Status	*KRAS* mRNA Expression Level in CTC Fraction				*CTLA-4* mRNA Expression Level in CTC Fraction			
	Low	High	Total	*p*-Value	Low	High	Total	*p*-Value
*KRAS* WT	4	3	7	0.56	4	3	7	0.24
*KRAS* MT	8	8	16		5	11	16	

WT: wild type, MT: mutant.

**Table 4 genes-14-01808-t004:** The correlations of *KRAS* mutation with clinicopathological features in patients with colorectal carcinoma (CRC).

Characteristics	CTCs			Cancer Tissue		
	MT (%)	WT (%)	*p*-Value	MT (%)	WT (%)	*p*-Value
**Gender**						
Female	1 (7.7%)	12 (92.3%)	0.386	8 (61.5%)	5 (38.5%)	0.54
Male	2 (20.0%)	8 (80.0%)		7 (70.0%)	3 (30.0%)	
**Age**						
≤60 years	1 (14.3%)	6 (85.7%)	0.684	3 (42.9%)	4 (57.1%)	0.14
>60 years	2 (12.5%)	14 (87.5%)		12 (75.0%)	4 (25.0%)	
**Size**						
≤40 mm	2 (15.4%)	11 (84.6%)	0.602	8 (61.5%)	5 (38.5%)	0.51
>40 mm	1 (10.0%)	9 (90.0%)		7 (70.0%)	3 (30.0%)	
**Site**						
Colon	2 (15.4%)	11 (84.6%)	0.602	8 (61.5%)	5 (38.5%)	0.51
Rectum	1 (10.0%)	9 (90.0%)		7 (70.0%)	3 (30.0%)	
**Grade**						
Well (1)	0 (0%)	4 (100%)	**0.039**	2 (50%)	2 (50.0%)	0.79
Moderate (2)	1 (6.25%)	15 (93.8%)		11 (68.8%)	5 (31.3%)	
Poor (3)	2 (66.7%)	1 (33.3%)		2 (66.7%)	1 (33.3%)	
**T Stage**						
I or II	0 (0%)	8 (100%)	0.094	4 (50.0%)	4 (50.0%)	0.09
III or IV	3 (20.0%)	12 (80.0%)		11 (73.3%)	4 (26.7%)	
**Lymph node status**						
Negative	0 (0%)	12 (100%)	**0.027**	7 (58.3%)	5 (41.7%)	0.39
Positive	3 (27.3%)	8 (72.7%)		8 (72.7%)	3 (27.3%)	
**Distant metastasis**						
Negative	3 (14.3%)	18 (85.7%)	0.444	13 (61.9%)	8 (38.1%)	0.18
Positive	0 (0%)	2 (100%)		2 (100%)	0 (0%)	
**Overall pathological stage**						
I or II	0 (0%)	12 (100%)	**0.027**	7 (58.3%)	5 (41.7%)	0.39
III or IV	3 (27.3%)	8 (72.7%)		8 (72.7%)	3 (27.3%)	
**MSI**						
Stable	3 (15.8%)	16 (84.2%)	0.266	14 (73.7%)	5 (26.3%)	0.06
High	0 (0%)	4 (100%)		1 (25.0%)	3 (75.0%)	
**Perineural invasion**						
Negative	1 (5.0%)	19 (95.0%)	**0.014**	12 (60.0%)	8 (40.0%)	0.09
Positive	2 (66.7%)	1 (33.3%)		3 (100%)	0 (0%)	
**Lymphovascular invasion**						
Negative	1 (6.7%)	14 (93.3%)	0.226	8 (53.3%)	7 (46.7%)	0.089
Positive	2 (25.0%)	6 (75.0%)		7 (87.5%)	1 (12.5%)	

MT, mutation; WT, wild type; Bold = *p*-value is significant.

**Table 5 genes-14-01808-t005:** The correlations of *KRAS* gene expression levels in CTCs with clinicopathological features of patients with colorectal carcinoma (CRC).

Characteristics	Total (23)	Low	High	*p*-Value
**Gender**				
Female	13 (56.5%)	6 (46.2%)	7 (53.8%)	0.552
Male	10 (43.5%)	4 (40.0%)	6 (60.0%)	
**Age**				
≤60 years	7 (30.4%)	1 (14.3%)	6 (85.7%)	**0.05**
>60 years	16 (69.6%)	9 (56.3%)	7 (43.8%)	
**Size**				
≤40 mm	13 (56.5%)	5 (53.8%)	8 (46.2%)	0.448
>40 mm	10 (43.5%)	5 (50.0%)	5 (50.0%)	
**Site**				
Colon	13 (56.5%)	7 (53.8%)	6 (46.2%)	0.237
Rectum	10 (43.5%)	3 (30.0%)	7 (70.0%)	
**Grade**				
Well (1)	4 (17.4%)	1 (25.0%)	3 (75.0%)	0.608
Moderate (2)	16 (69.6%)	8 (50.0%)	8 (50.0%)	
Poor (3)	3 (13.04%)	1 (33.3%)	2 (66.7%)	
**T Stage**				
I or II	8 (34.8%)	4 (50.0%)	4 (50.0%)	0.490
III or IV	15 (65.2%)	6 (40.0%)	9 (60.0%)	
**Lymph node status**				
Negative	12 (52.2%)	7 (58.3%)	5 (41.7%)	0.129
Positive	11 (47.8%)	3 (27.3%)	8 (72.75)	
**Distant metastasis**				
Negative	21 (91.3%)	9 (42.9%)	12 (57.1%)	0.692
Positive	2 (8.7%)	1 (50.0%)	1 (50.0%)	
**Overall pathological stage**				
I or II	12 (52.2%)	7 (58.3%)	5 (41.7%)	0.129
III or IV	11 (47.8%)	3 (27.3%)	8 (72.7%)	
**Microsatellite instability (MSI)**				
Stable	19 (82.6%)	9 (47.4%)	10 (52.6%)	0.401
High	4 (17.4%)	1 (25.0%)	3 (75.0%)	

Bold = *p*-value is significant.

**Table 6 genes-14-01808-t006:** The correlations of *CTLA-4* gene expression levels in CTCs with clinicopathological features of patients with colorectal carcinoma (CRC).

Characteristics	Total (23)	Low	High	*p*-Value
**Gender**				
Female	13 (56.5%)	4 (30.8%)	9 (69.2%)	0.306
Male	10 (43.5%)	5 (50.0%)	5 (50.0%)	
**Age**				
≤60 years	7 (30.4%)	3 (42.9%)	4 (57.1%)	0.582
>60 years	16 (69.6%)	6 (37.5%)	10 (62.5%)	
**Size**				
≤40 mm	13 (56.5%)	6 (46.2%)	7 (53.8%)	0.363
>40 mm	10 (43.5%)	3 (30%)	7 (70%)	
**Site**				
Colon	13 (56.5%)	6 (46.2%)	7 (53.8%)	0.363
Rectum	10 (43.5%)	3 (30%)	7 (70%)	
**Grade**				
Well (1)	4 (17.4%)	4 (100%)	0 (0%)	**0.011**
Moderate (2)	16 (69.6%)	4 (25.0%)	12 (75.0%)	
Poor (3)	3 (13.0%)	1 (33.3%)	2 (66.7%)	
**T Stage**				
I or II	8 (34.7%)	5 (62.5%)	3 (37.5%)	**0.036**
III or IV	15 (65.2%)	4 (26.7%)	11 (73.3%)	
**Lymph node status**				
Negative	12 (52.2%)	6 (50.0%)	6 (50%)	0.247
Positive	11 (47.8%)	3 (27.3%)	8 (72.7%)	
**Distant metastasis**				
Negative	21 (91.3%)	9 (42.9%)	12 (57.1%)	0.147
Positive	2 (8.7%)	0 (0%)	2 (100%)	
**Overall pathological stage**				
I or II	12 (52.2%)	6 (50.0%)	6 (50.0%)	0.247
III or IV	11 (47.8%)	3 (27.3%)	8 (72.7%)	
**Microsatellite instability (MSI)**				
Stable	19 (82.6%)	6 (31.6%)	13 (68.4%)	0.107
High	4 (17.4%)	3 (75%)	1 (25%)	

Bold = *p*-value is significant.

## Data Availability

The data presented in this study are available in Correlation between *KRAS* Mutation and *CTLA-4* mRNA Expression in Circulating Tumour Cells: Clinical Implications in Colorectal Cancer.

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
