# Peer review of "Correlation between KRAS Mutation and CTLA-4 mRNA Expression in Circulating Tumour Cells: Clinical Implications in Colorectal Cancer"

_genes, 2023, doi:10.3390/genes14091808_

Round 1

Reviewer 1 Report

This research was conducted in a logical manner. This study investigated the genomic variations of KRAS in CTCs and matched CRC tissues and compared mRNA expression of KRAS and CTLA-4 between wild-type and KRAS-mutated CTCs and CRC tissues. The results revealed that CTCs harboring the KRAS variant were different from matched CRC tissue. Patients harboring KRAS mutation in CTCs were more likely to have poorly differentiated tumor with lymph node metastasis and perineural invasion. Though insignificant, CTLA-4 mRNA expression was higher in patients carrying KRAS mutation. These findings imply the genetic basis of KRAS with immunotherapeutic target molecules.

However, most of the results did not show a statistical difference. More importantly, the significance of the study was not clearly stated and the results were inadequate. The authors should carefully revise the article.

1. As the result showed in 3.2: There was no concordance between KRAS mutations in CTCs and primary tumors, and CTCs harboring the KRAS variant were different from matched CRC tissue. So the appropriate treatment regimen should be chosen based on the mutational status of the in primary tumor or the CTCs in clinical treatment? And the authors should elaborate on the importance of CTCs in clinical care, which will help the reader to understand it better..

2. Why did the authors test CTLA-4 expression levels and is there any potential link between CTLA-4 and KRAS mutations in cancer? The author should elaborate this in Introduction section.

3. Result 3.3: The authors did not find any correlation of KRAS mutation with the KRAS and CTLA-4 mRNA expression in CTCs and primary tumors. Perhaps the authors could try to test the protein content of KRAS mutants.

4. Most of the data in this paper did not show positive results, which may be related to the limited sample size (n=23), and it is recommended that the sample size be expanded and re-analyzed.

5. There are some spelling errors and grammatical errors in the article. Please check again and correct.

6. Line 65, 202, 203: Letter sizes are inconsistent.

There are some spelling errors and grammatical errors in the article.

Author Response

  1. Thank you for your suggestions. Yes, there was no concordance between KRAS mutations in CTCs and primary tumours. Moreover, CTCs harbouring the KRAS variant were different from matched CRC tissue. The information regarding genotype shifting from primary tumours to CTCs might be useful in clinical decision-making. Hence, genetic profiling of CTCs may provide an innovative approach for understanding the dynamic characteristics of tumours in real-time, as an alternative to invasive procedures. We have explained why the molecular characterisation of CTCs is important in precision medicine in the Introduction section (Section 1: p.2, Line 70-97).

  1. Several reports found that genetic alterations can change the expression of immune checkpoint molecules such as PD-L1, and CD47 (Section 1: p. 2, lines 63-69). Additionally, our previous study found a positive correlation between the higher expression of KRAS and CTLA-4 which suggests that there might be a correlation between the CTLA-4 gene expression level and KRAS mutation. Hence, we aimed to explore the gene expression level in KRAS wild-type and KRAS mutant CRC cases. We have elaborated the explanation in the Introduction section (Section 1: p.2, Line 88-97).

  1. Yes, we did not find any significant correlation between KRAS mutation with the KRAS and CTLA-4 mRNA expression in CTCs and primary tumours. This may be due to the low number of sample sizes for gene expression analysis. Another key factor might be the extremely heterogeneous nature of cancer cells between individual patients. However, the tendency to have higher CTLA-4 expression in KRAS mutant cases might suggest that there might be a correlation.

Due to the low number of cells, the protein expression analysis is not feasible to perform on CTC samples. As this study investigated the clinical potential of the molecular characterisation of CTCs, expression profiling of the target genes in mRNA levels of the samples was carried out.  Like previous studies working on CTCs [1-3], herein we used mRNA expression levels in KRAS wild-type and mutant cases. This information is added to the Result section 3.3.

References

  1. Steinert G, Schölch S, Niemietz T, et al. Immune escape and survival mechanisms in circulating tumor cells of colorectal cancer. Cancer Res. 2014;74(6):1694-1704.
  2. Hensler M, Vančurová I, Becht E, et al. Gene expression profiling of circulating tumor cells and peripheral blood mononuclear cells from breast cancer patients. Oncoimmunology. 2015;5(4):e1102827-e1102827. doi:10.1080/2162402X.2015.1102827
  3. Chikamatsu K, Tada H, Takahashi H, et al. Expression of immune-regulatory molecules in circulating tumor cells derived from patients with head and neck squamous cell carcinoma. Oral Oncology. 2019/02/01/ 2019;89:34-39. doi:https://doi.org/10.1016/j.oraloncology.2018.12.002

  1. The issue of insufficient sample size and concerns regarding statistical significance were also noted. During this study, we have collected 57 samples which is a substantial number based on the recent publications. Among these samples, some (n=8) cannot be used for the whole population genomic screening due to the low yield of cells and poor mRNA quality. Moreover, gene expression analysis and genomic profiling were only performed on CTC-positive patient samples (n=23).

  1. Spelling errors and grammatical errors have been thoroughly checked and corrected.

  1. Corrected, thanks.

Reviewer 2 Report

In this study, Aktar et al. and their colleagues used patient specimens to explore the correlation between KRAS mutations and CTLA-4 mRNA expression in circulating tumor cells (CTCs). They found that patients with KRAS-mutated CTCs exhibited poorly differentiated tumors, lymph node metastasis, and perineural invasion. Additionally, although not statistically significant, the mRNA expression of CRLA-4 was relatively higher in these cases. This paper is presented in a straightforward manner, primarily focused on description. It offers preliminary insights into the potential used of CTC genotyping to assess immunotherapy, which could prove beneficial for more in-depth studies in the future. I don't have additional scientific questions to ask. 

Author Response

Thank you for the positive comments and with nothing need to edit. 
